# Molecular and Functional Characterization of a Short-Type Peptidoglycan Recognition Protein, Ct-PGRP-S1 in the Giant Triton Snail *Charonia tritonis*

**DOI:** 10.3390/ijms231911062

**Published:** 2022-09-21

**Authors:** Wenguang Liu, Bing Liu, Gege Zhang, Huixia Jia, Yang Zhang, Xitong Cen, Gaoyou Yao, Maoxian He

**Affiliations:** 1CAS Key Laboratory of Tropical Marine Bio-resources and Ecology, Guangdong Provincial Key Laboratory of Applied Marine Biology, South China Sea Institute of Oceanology, Chinese Academy of Sciences, Guangzhou 510301, China; 2Southern Marine Science and Engineering Guangdong Laboratory (Guangzhou), Guangzhou 511458, China; 3University of Chinese Academy of Sciences, Beijing 100049, China

**Keywords:** *Charonia tritonis*, peptidoglycan recognition protein, PGN, amidase activity, antibacterial activity

## Abstract

Peptidoglycan recognition proteins (PGRPs) are a family of pattern recognition receptors (PRRs) involved in host antibacterial responses, and their functions have been characterized in most invertebrate and vertebrate animals. However, little information is available regarding the potential function of PGRPs in the giant triton snail *Charonia tritonis*. In this study, a short-type PGRP gene (termed *Ct-PGRP-S1*) was identified in *C. tritonis*. Ct-PGRP-S1 was predicted to contain several structural features known in PGRPs, including a typical PGRP domain (Amidase_2) and Src homology-3 (SH3) domain. The *Ct-PGRP-S1* gene was constitutively expressed in all tissues examined except in proboscis, with the highest expression level observed in the liver. As a typical PRR, Ct-PGRP-S1 has an ability to degrade peptidoglycan (PGN) and was proven to have non-Zn^2+^-dependent amidase activity and antibacterial activity against *Vibrio*
*alginolyticus* and *Staphylococcus aureus*. It is the first report to reveal the peptidoglycan recognition protein in *C. tritonis*, and these results suggest that peptidoglycan recognition protein Ct-PGRP-S1 is an important effector of *C. tritonis* that modulates bacterial infection resistance of *V. alginolyticus* and *S. aureus*, and this study may provide crucial basic data for the understanding of an innate immunity system of *C. tritonis*.

## 1. Introduction

As we know, invertebrates rely solely on innate immunity to protect themselves against various microorganisms because they lack an acquired immune system as vertebrates have. Innate immunity employs several kinds of pattern recognition receptors (PRRs) to discriminate self from non-self, and initiate subsequent immune cascades to eliminate harmful invaders [1]. Peptidoglycan recognition proteins (PGRPs), one of the PRRs, specifically recognize bacterial peptidoglycan (PGN) [2], which is a unique component from the cytoderm of prokaryotic organisms, which helps to maintain the integrity of the microbial cells [3].

Since PGRP was purified and shown to bind specifically to bacterial PGN, activating an antibacterial response [4], subsequently, a large amount of PGRP homologues have been identified from both invertebrates and vertebrates. Usually, each species contains more than one homologue of PGRP, such as 13 genes from Drosophila [5,6,7], 4 genes from mammals [8] and 7 genes from Pacific oyster *Crassostrea gigas* [9]. To date, there have been over 100 PGRPs found in all species and expressed in various tissues, especially in immune tissues [10,11]. According to their structure and molecular weight, PGRPs are divided into three classes: short and extracellular (20–25 kDa, PGRP-S), intermediate (PGRP-I) with two transmembrane domains, and long and intercellular (>90 kDa, PGRP-L) types [1,12,13]. The diversity of PGRPs in structure and classification indicates that PGRPs perform multiple functions. Generally, their functions are divided into three categories: PRRs, signal regulation molecules, and effectors, all of which are involved in the antimicrobial innate immunity process [14], indicating their prominent roles in host innate immune defense.

In recent years, studies focusing on marine PGRPs have made great progress. These marine PGRPs were proven to act as PRRs, opsonins, and bactericidal amidases similarly to their counterparts in terrestrial vertebrates [1]. For example, some marine PGRPs serve as PRRs by specifically binding PGN and/or bacteria [11,15,16,17], while others enhance agglutination or phagocytosis [1,18,19,20]. Moreover, many marine PGRPs exhibit amidase activity and degrade PGN, thereby suppressing bacterial growth [18,19]. These important functions strongly indicate crucial roles in the immune system of marine organisms in the defense against potential pathogens. Thus far, PGRPs have been identified from only a few species, including amphioxus [19], scallop [18], razor clam [21], abalone [22], oyster [9], sea star [23], red drum [24], vent mussel [25], squid [26], sea cucumber [27] and tongue sole [20,28].

*C. tritonis* is an endangered gastropod species of its ecological and economic importance [29]. Protection of *C. tritonis* populations requires solid knowledge of biology and ecology, which can be further improved by studies on maintenance and breeding in captivity [30]. However, artificial cultures will inevitably be affected by bacteria-induced diseases. Given the important roles that PGRPs play in the recognition and inhibition of bacterial pathogens and the subsequent initiation of immune pathways in response to pathogen invasion, exploring novel family members and investigating the functional characteristics in *C. tritonis* are essential for understanding the evolution, diversity, and immune mechanisms of PGRPs as well as exploring effective strategies and efficacious immunomodulatory methods to control bacterial diseases in this important species. However, compared with the remarkable progress in insects and mammals or other marine animals, there is a research gap on the studies of *C. tritonis* PGRPs.

Here, we identified a PGRP gene from the transcriptome of the giant triton snail (*Charonia tritonis*). Given that *C. tritonis* is an endangered gastropod species, we did not perform pathogens challenge assay to investigate which play the most vital roles in antibacterial immune response in *C. tritonis*. To further explore its characterization and potential functions, we obtained one full-length open reading frame of the cDNA sequence of peptidoglycan recognition protein Ct-PGRP-S1 and determined its mRNA expression patterns of *Ct-PGRP-S1* in different tissues of *C. tritonis*. The recombinant Ct-PGRP-S1 protein was expressed, and its functions were characterized including amidase activity and antibacterial activity. It is the first report to reveal the peptidoglycan recognition protein in *C. tritonis*. These results suggest that peptidoglycan recognition protein Ct-PGRP-S1 is an important effector of *C. tritonis* that modulates bacterial infection resistance of *V.*
*alginolyticus* and *S. aureus*, and this study may provide crucial basic data for the understanding of an innate immunity system of *C. tritonis*.

## 2. Results

### 2.1. Identification of Ct-PGRP-S1 cDNA Sequence from Transcriptome of C. tritonis

Based on the gene annotation of the transcriptome of *C. tritonis* (NCBI sample ID: PRJNA695322) [29], we obtained the cDNA sequence of *Ct-PGRP-S1* (ID: Unigene31272_All), which was annotated as peptidoglycan recognition protein. The cDNA sequence of the open reading frame of *Ct-PGRP-S1* was then determined by TA-cloning and deposited in Genbank under the accession number OP351736. As shown in Figure 1, the open reading frame (ORF) of *Ct-PGRP-S1* is 867 bp in size, which encodes for a 288 a.a. protein precursor with a calculated molecular weight (MW) of 31.42 KDa.

### 2.2. Molecular Characterization and Bioinformatic Analyses of Ct-PGRP-S1

Based on the SignalP program, Ct-PGRP-S1 was predicted to contain a 22 a.a. signal peptide (Figure 1). SMART program analysis revealed that the putative mature protein of *Ct-PGRP-S1* contained an Amidase_2 domain and a SH3_3 domain (Figure 1A,B). In addition, a 3-D model of the Ct-PGRP-S1 protein precursor was predicted by the SWISS-MODEL server (Figure 1C), which shared 47.06% similarity with the crystal structure of peptidoglycan recognition protein 3 of *Branchiostoma belcheri tsingtauense*.

Blast analysis showed that the deduced amino acid sequences of *Ct-PGRP-S1* share 63.53% and 62.26% identity with the previously identified sequence from *Littorina littorea* (UFA46008.1) and *Pomacea canaliculate* (XP_025096405.1). Furthermore, a range of PGRP proteins from various gastropod mollusks was collected to study the evolutionary relationships by phylogenetic analysis. The results of the phylogenetic analysis revealed that our newly identified Ct-PGRP-S1 has the shortest evolutionary distance from the evolutionary distance from *Littorina littorea* and *Batillaria attramentaria* (Figure 2A). Multiple alignments of amino acid sequences of PGRP-S (Figure 2B) showed that PGRPs were relatively conserved among these species, especially C-terminal, which belongs to the typical PGRP domain. Furthermore, four Zn^2+^-binding amino acids (two histidines, one tyrosine, and one cysteine) [31,32,33], which are essential for amidase activity of PGRPs, were conserved among these species.

### 2.3. Expression Profiles of Ct-PGRP-S1 mRNA in Different Tissues

The mRNA expression pattern of *Ct-PGRP-S1* was analyzed in various tissues of *C. tritonis* by qPCR. As shown in Figure 3A, the expression of *Ct-PGRP-S1* was ubiquitously detected in almost all selected tissues except in proboscis, with the strongest expression level in the liver, followed by the mantle, tentacle and salivary glands.

In addition, the transcriptome data before and after feeding showed that the highest expression level of *Ct-PGRP-S1* was seen in the liver among three tissues (digestive glands, liver, salivary glands), followed by the digestive glands and salivary glands. However, the RPKM (reads per kilobase per million mapped reads) of *Ct-PGRP-S1* after feeding was upregulated significantly by more than 150-fold in the liver of *C. tritonis* but remained unchanged in digestive glands and salivary glands. It implied that feeding increases the expression of *Ct-PGRP-S1* in the liver but has on impact on the transcript abundance of *Ct-PGRP-S1* in digestive glands and salivary glands (Figure 3B).

### 2.4. Prokaryotic Expression Vector Construction and Purification of rCt-PGRP-S1

The ORF sequence of *Ct-PGRP-S1* was obtained from *C. tritonis* cDNA and was cloned into the pET-28b vector to produce rCt-PGRP-S1 in *E. coli* BL21(DE3). rCt-PGRP-S1 was overexpressed after IPTG induction and subsequently purified by using Ni-NTA beads. SDS-PAGE analysis showed that the purified rCt-PGRP-S1 band was approximately 30 kDa (Figure 4A), which was in accordance with the Western blot band detected by the anti-6×His Tag antibody in Figure 4B.

### 2.5. Amidase Activity and Antibacterial Activity of rCt-PGRP-S1 Protein

Based on the domain prediction (Figure 1A,B) and high sequence similarity of Ct-PGRP-S1 to PGRP-S in other animals (Figure 2B), we speculated that Ct-PGRP-S1 could have amidase activity. Considering the important role of amidase activity against pathogens [14], we further measured the release of muramic acid from the interpeptide chains of *S. aureus* PGN. When PGN was degraded by rCt-PGRP-S1 in Tris-ZnCl_2_ buffer, the OD_540nm_ value decreased dramatically from 0.425 to 0.266 within 120 min (Figure 5). However, a much similar level of activity was also detected in the group rCt-PGRP-S1 without Zn^2+^ as evidenced by the dramatically decreased OD_540nm_ value (from 0.392 to 0.261), indicating that the amidase activity of rCt-PGRP-S1 was not Zn^2+^ dependent. This feature is inconsistent with previous results [16,21], which demonstrated that the amidase activity toward PGN of PGRP-S must be Zn^2+^ dependent. The result here suggests that rCt-PGRP-S1 has amidase activity toward PGN, but the amidase activity was not Zn^2+^ dependent.

The inhibition effects of rCt-PGRP-S1 were investigated on the growth of *V. alginolyticus* (Figure 6A), *V.*
*cholerae* (Figure 6B), *V. parahaemolyticus* (Figure 6C), *E. coli* (Figure 6D), and *S. aureus* (Figure 6E). In the presence of Zn^2+^, no significantly different growth curves of *E. coli*, *V. cholerae* and *V. parahaemolyticus* were exhibited, implying that rCt-PGRP-S1 showed no apparent inhibition effects on *E. coli*, *V. cholerae* and *V. parahaemolyticus*. By contrast, rCt-PGRP-S1 weakly inhibited the growth of *V. alginolyticus,* as they showed similar lag phase but had lower maximum cell density. In addition, *S. aureus* exhibited a prolonged lag phase (>2 h) compared with the group without rCt-PGRP-S1, indicating that rCt-PGRP-S1 suppresses the growth of *S. aureus* bacteria.

## 3. Discussion

PGRPs are a family of PRRs that play crucial roles in host innate immune responses [14,34,35], and their functions have been characterized in most invertebrates and vertebrates. However, little information is available to date regarding the functions of *C. tritonis* PGRPs. In the present study, we identified a PGRP gene named as *Ct-PGRP-S1* in the endangered species *C. tritonis*. The predicted protein of Ct-PGRP-S1 possesses a typical PGRP domain (Amidase_2 domain) [14,35,36] and Src homology-3 (SH3) domain. Ct-PGRP-S1 and other short-type PGRPs in gastropods all contained a conserved C-terminal typical PGRP domain, indicating that Ct-PGRP-S1 is a kind of short-type PGRP in gastropods.

As we know, most PGRPs contain one C-terminal PGRP domain (approximate 165 amino acids), which is homologous to bacteriophage and bacterial type 2 amidases [5,8,12]. However, not all PGRPs contain a SH3 domain. We performed domain prediction on all the proteins described in the phylogenetic tree, and we found that only a few PGRP-S proteins contained the SH3_3 domain, including *P. canaliculate* (XP_025096405.1), *D. polymorpha* (KAH3862486.1), and *M. mercenaria* (XP_045194511.1). Src homology-3 (SH3) domains, consisting of approximately 60 amino acids, mediate intramolecular associations and form supramodules with their own supertertiary structures [37], which further regulate a wide spectrum of cellular processes such as cell–environment and cell–cell communication, cytoskeletal rearrangement, cell migration and growth, protein trafficking and degradation, and immune responses [38,39,40]. Bacterial homologs of the SH3 domain were primarily extracellular domains in periplasmic or cell-wall-associated proteins [41,42,43]. Tamai et al. showed that Src homology 3 domains are indispensable for the lytic activity of endolysin as the cell wall-binding domain and suggested that the SH3 domain presumably recognizes peptide side-chains to fix the catalytic domain in proper orientation for binding to the glycan backbone [44]. Therefore, the SH3 domain of Ct-PGRP-S1 possibly facilitates PGRP to interact with peptidoglycans on the cell wall and exerts immune inhibition.

One common feature between invertebrate and vertebrate PGRPs is amidase activity, which hydrolyses the amide bond between MurNAc and L-alanine in bacterial PGN [35]. Previous studies suggest that amidase-active PGRPs must contain four conserved Zn^2+^-binding residues, including two histidines, one tyrosine, and one cysteine [31,32,33]. Here, Ct-PGRP-S1 has these four residues (histidine: +146 a.a. and +255 a.a.; tyrosine: +208 a.a.; cysteine: +263 a.a.), indicating that Ct-PGRP-S1 is a kind of amidase-active PGRP. However, the amidase activity of recombinant Ct-PGRP-S1 has been proven to be non-Zn^2+^-dependent. This result was unexpected because the three Zn^2+^-binding residues in the PGRP domain were completely conserved in Ct-PGRP-S1. However, this is not an exception. Jaeeun et al. demonstrated that the absence or presence of metal ions, such as EDTA, Mg^2+^, Ca^2+^, Zn^2+^, or Mn^2+^, did not affect the lytic activity of endolysin, although it has three completely conserved Zn^2+^-binding residues in the PGRP domain [45], which is the same as the case of Ct-PGRP-S1.

*C. tritonis* are specialist predators of *Acanthaster planci* (crown-of-thorns starfish), which is important for the biodiversity of coral reefs and their biological communities [29]. However, *C. tritonis* are considered vulnerable, and a decline in their populations will contribute to recurring population outbreaks of *A. planci*. Aquaculture has been considered as one effective approach that could help restore natural populations of *C. tritonis* and mitigate coral loss [46]. We could speculate that in the process of aquaculture, they may be at risk of bacterial infection, especially the *Vibrio,* since there are many reports of bacterial infection causing a massive economic loss in the aquaculture industry [47,48,49,50]. Meanwhile, various aquatic animals have suffered from *S. aureus* infections [51,52,53,54,55]. In the present study, co-incubation assay revealed that recombinant Ct-PGRP-S1 was proven to have antibacterial activity against *V. alginolyticus* and *S. aureus* by direct contact, reflecting the importance of Ct-PGRP-S1 in the antibacterial defense of *C. tritonis* and providing important evidence to suggest that amidase-active PGRPs may function as the key effectors to promote antibacterial response in hosts.

In summary, a gene coding a short-type PGRP, named *Ct-PGRP-S1*, was identified in *C. tritonis.* Except in proboscis, the *Ct-PGRP-S1* gene was constitutively expressed in all tissues examined, with the highest expression level was observed in liver. In addition, the recombinant Ct-PGRP-S1 has been proven to have PGN-degrading activity, non-Zn^2+^-dependent amidase activity and antibacterial activity against *V. alginolyticus* and *S. aureus*. To the best of our knowledge, the present study represents the first discovery on the function of PGRPs in *C. tritonis*, thus contributing to a better understanding of the functional evolution of PGRPs in gastropoda.

## 4. Materials and Methods

### 4.1. Animals and Tissue Collection

The adult *C. tritonis* were obtained from Nansha archipelagic waters of the South China Sea. Tissue samples (salivary glands, tentacle, proboscis, mantle and liver) were collected and quickly frozen in liquid nitrogen for 24 h, and then stored at 80 °C until total RNA extraction as previous described by Zhang et al. [29]. The animal experiments were conducted following the guidelines and approval of the Ethics Committees of the South China Sea Institute of Oceanology, Chinese Academy of Sciences (approval code: 2022-001; approval date: 25 July 2022).

### 4.2. Molecular Cloning and Bioinformatics Analysis of Ct-PGRP-S1

A full-length cDNA sequence of the *C. tritonis Ct-PGRP-S1* gene was found in a transcriptomic database constructed by our laboratory (NCBI sample ID: PRJNA695322, accessed on 27 January 2021), and the open reading frame (ORF) sequence of *Ct-PGRP-S1* was obtained using the primers annotated *Ct-PGRP-S1*-F/R (Table 1). The sequence was deposited in GenBank (GenBank ID: OP351736, accessed on 1 September 2022). The cDNA extraction of total RNA and reverse transcription of first-strand cDNA were performed following the procedure described by Liu et al. [56]. Prediction of the structural domains of the Ct-PGRP-S1 protein was conducted with the SMART program, and the three-dimensional (3-D) model was generated by using the SWISS-MODEL server. The amino acid sequence alignment was performed with clustalx1.8, and the phylogenetic tree was built by neighbor-joining method with 1000 bootstrap replicates using MEGA 6.0. The species used in this analysis and their accession ID are shown as follows: *Littorina littorea* UFA46008.1, *Pomacea canaliculate* XP_025096405.1, *Dreissena polymorpha* KAH3862486.1, *Mercenaria mercenaria* XP_045194511.1, *Reishia clavigera* AET43945.1, *Ostrea edulis* XP_048750675.1, *Crassostrea virginica* XP_022311450.1, *Mytilus edulis* CAG2232607.1, *Mytilus galloprovincialis* AJQ21531.1, *Tegillarca granosa* QZM06958.1, *Crassostrea gigas* XP_011420323.2, *Batillaria attramentaria* KAG5698050.1, *Mizuhopecten yessoensis* XP_021376412.1, *Gigantopelta aegis* XP_041366403.1, *Pecten maximus* XP_033761487.1, *Lottia gigantea* XP_009064695.1, *Bulinus truncatus* KAH9512611.1, *Physella acuta* AEH26026.1.

### 4.3. RNA Isolation and Quantitative Reverse Transcription PCR (qRT-PCR) Analysis

Total RNA of multiple tissues, including salivary glands, tentacle, proboscis, mantle and liver from adult *C. tritonis*, was isolated using the TransZol Up Plus RNA Kit (TransGen Biotech, Beijing, China). Prime-Script RT Kit with gDNA Eraser (Takara Bio Inc., Kusatsu City, Japan) was used for reverse transcription. TB Green^®^ Premix Ex Taq™ II (Takara Bio Inc., Kusatsu City, Japan) was used for quantitative real-time PCR detection. Tissue expression pattern of *Ct-PGRP-S1* was detected by quantitative real-time PCR (qPCR) using the gene-specific primers (Table 1), and 18s rDNA was used as an internal reference. A Thermal Cycler Dice Real Time System III (Takara Bio Inc.) instrument was used for qPCR with the following parameters: for the holding stage, 95 °C for 30 s; for the two step PCR stage, 95 °C for 5 s and 60 °C for 30 s (repeated 40 times) with the fluorescence recorded at 60 °C; and for the melting curve stage, 90 °C for 15 s, 60 °C for 30 s, and then 95 °C for 15 s with the fluorescence recorded every 0.05 s. The relative expression levels were calculated using the threshold cycle (ΔΔCT) method [57]. Measurements were performed in triplicate biologically and technically. Statistical significance was determined by the Student’s *t* test (ns *p* > 0.05, * *p* < 0.05, ** *p* < 0.01).

### 4.4. Construction of Prokaryotic Expression Vector

The cDNA segment (67–864 bp) including the full-length coding sequence of the *Ct-PGRP-S1* gene except for the sequence of predicted signal peptide was amplified by PCR with the primer sets Ct-PGRP-S1-orf-F/R (Table 1). The plasmid pET28b was amplified with linearized primer pairs pET28b-F/R, and the fragments were then inserted into plasmid pET28b with a ClonExpress^®^ II One Step Cloning Kit (Vazyme Biotech Co., Ltd., Nanjing, China) to obtain a recombinant plasmid, which was transformed into *E. coli* BL21 (DE3)-competent cells. Subsequently, PCR and sequencing were used to check for the presence of the target genes with the primer pair pET28b-check-F/R.

### 4.5. Over-Expression and Purification of Recombinant Ct-PGRP-S1 Protein

Bacterial Strain rCt-PGRP-S1-pET28b/BL21 (DE3) was grown overnight in LB medium plus 50 μg kanamycin at 30 °C with shaking at 200 rpm. Then, 500 μL of overnight *E. coli* BL21 (DE3) culture was diluted into 500 mL fresh Luria broth (LB) with 50 mg/mL kanamycin (1:1000). The culture was incubated at 37 °C with shaking at 200 rpm until the cell density reached 0.6 at 600 nm optical density. Overexpression of recombinant proteins was induced by isopropyl-β-D-thiogalactoside (IPTG). Briefly, the culture was shifted to 28 °C and induced by IPTG at 0.5 mM final concentration. After 4 h of induction, the induced cells were harvested by centrifugation at 6000× *g* for 3 min at 4 °C. The cells were resuspended in 10 mL lysis buffer (50 mM NaH_2_PO_4_, 300 mM NaCl, 5 mM imidazole, and 100 mM protease inhibitor PMSF). Pressure cell disruptor (Constant Systems, Daventry, UK) was used for cell disruption, and cell debris was pelleted by centrifugation at 15,000× *g* for 30 min at 4 °C. The supernatant was incubated with 2 mL of nickel agarose beads for 3 h with rotation. The slurry was eluted by using a 50–500 mM imidazole gradient. The reaction mixture was then passed through a nickel agarose column, and purified protein was concentrated using Amicon Ultra centrifuge tube (Millipore, Burlington, MA, USA) and stored in Tris buffer at −80 °C. The purified rCt-PGRP-S1 protein was characterized using 12% sodium dodecyl sulfate-polyacrylamide gel electrophoresis (SDS-PAGE). Western Blot was carried out according to the methods described by Liu et al. [58]. Briefly, the proteins were separated by SDS-PAGE, and transferred to 0.2 μm polyvinylidene difluoride (PVDF) membranes (Millipore, Burlington, MA, USA). The purified protein was detected using anti-6×His tag monoclonal antibody (Sangon Biotech, Shanghai, China), and SuperSignal™ West Pico PLUS (Thermo Fisher scientific, Waltham, MA, USA) was used for visualization.

### 4.6. Amidase Activity Assay

The enzyme activity of rCt-PGRP-S1 toward PGN from *S. aureus* (Macklin, Shanghai, China) was determined according to a previous method with minor modification [1]. In brief, a total of 40 μg PGN (1 mg/mL) was incubated with 50 μg rCt-PGRP-S1 protein in Tris buffer (20 mM Tris-HCl, 150 mM NaCl, pH 7.2) or Tris-ZnCl_2_ buffer (20 mM Tris-HCl, 150 mM NaCl, 10 mM ZnCl_2_, pH 7.2) at 37 °C. Simultaneously, the Tris buffer without PGN was used as control group. The absorbance (OD) values were recorded at 540 nm per 10 min lasting for 120 min.

### 4.7. Antimicrobial Activity Assay

The antimicrobial activity of purified recombinant *Ct-PGRP-S1* protein was investigated according to the method described by He et al. [59] with slight modification. In brief, the microbial strains used in the assays included Gram-negative bacteria *Escherichia coli* and *Vibrio alginolyticus*, *Vibrio cholerae, Vibrio parahaemolyticus* and Gram-positive bacteria *Staphylococcus aureus*, grown overnight in LB medium at 37 °C. All cells were grown to mid-logarithmic phase (OD_600nm_ ≈ 0.5). The bacterial growth rate method was used to investigate the antimicrobial activity of the purified recombinant Ct-PGRP-S1 protein. The purified recombinant protein was diluted with sterile Tris buffer (50 mM Tris-HCl, pH 8.0) to final concentrations of 50 μg/mL, and 50 μL of rCt-PGRP-S1 was added to 900 μL OD_600nm_ of 0.1 *E. coli*, *V. alginolyticus*, *V. cholerae, V. parahaemolyticus*, and *S. aureus*. Cultures (3 replicates in each case) were then incubated at 37 °C with continuous shaking at 200 rpm in 96-well plates. The wells in which the recombinant protein was replaced with Tris were used as the loading control. The wells in which the recombinant protein and the bacterial culture were left out were used as the blank control. OD_600nm_ was measured at regular time intervals using the Multiskan Ascent plate reader (Thermo Fisher Scientific, Waltham, MA, USA). The antimicrobial activity of Ct-PGRP-S1 was measured by the growth kinetics or maximum cell densities of differenced bacteria.

## Figures and Tables

**Figure 1 ijms-23-11062-f001:**
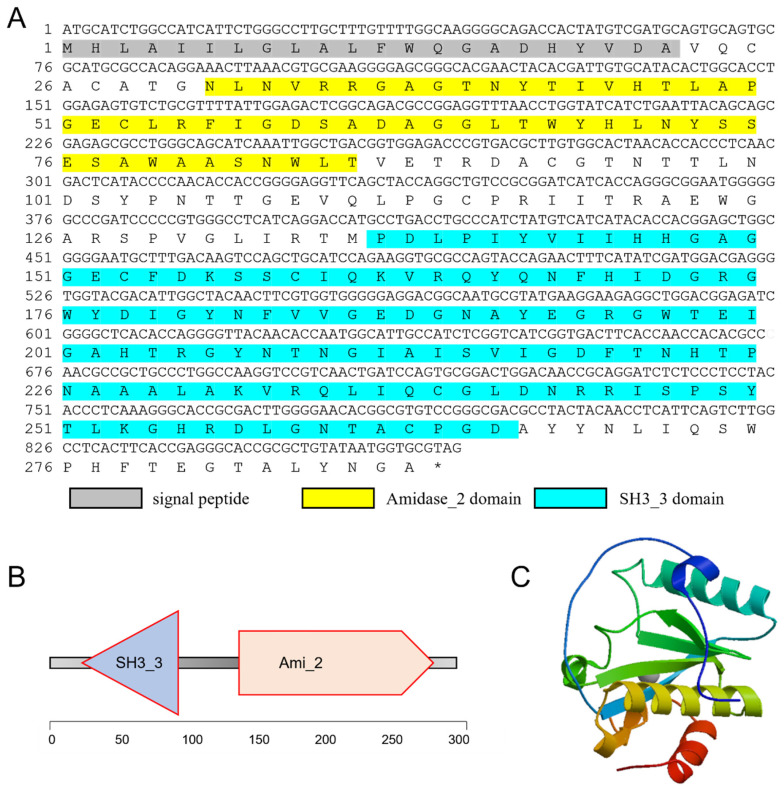
(**A**) The full-length cDNA sequence of *Ct-PGRP-S1* and its deduced amino acid sequence. The signal peptide is shadowed. The Amidase_2 (N-acetylmuramoyl-L-alanine amidase, PF01510.28) domain is yellow. The SH3_3 (Bacterial SH3 domain, PF08239.14) domain is blue. (**B**) Simplified structural domain model of Ct-PGRP-S1 predicted by SMART program. (**C**) Three-dimensional (3-D) protein model for Ct-PGRP-S1 predicted by SWISS-MODEL server.

**Figure 2 ijms-23-11062-f002:**
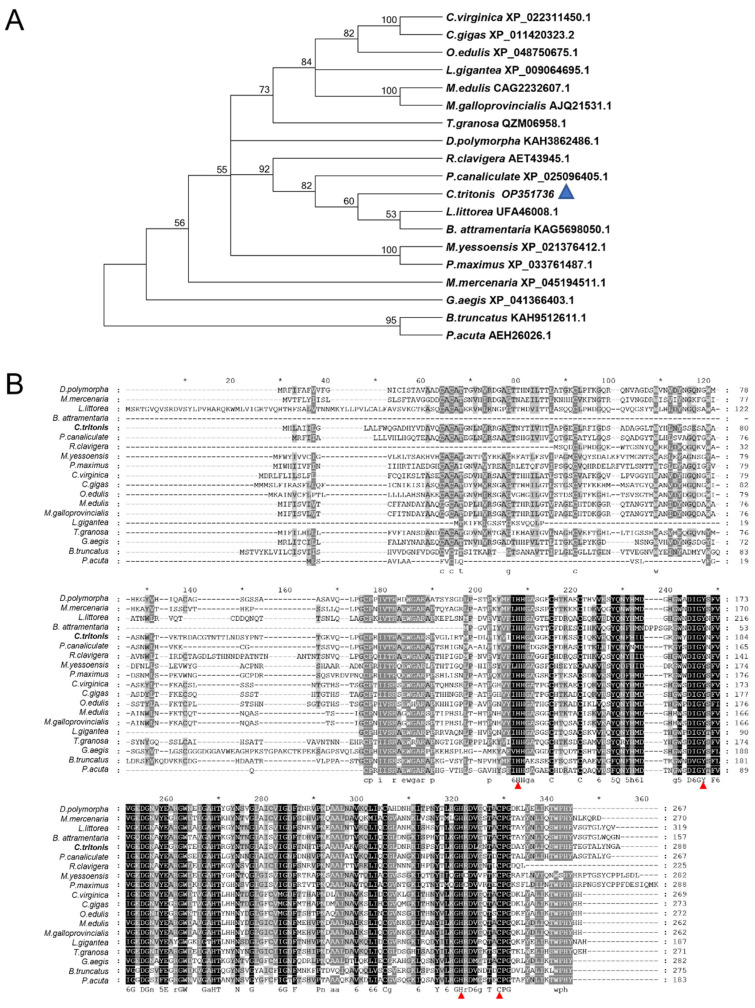
(**A**) Phylogenetic analysis of PGRP-S among various species of gastropod mollusk using the neighbor-joining method with a bootstrap value of 1000. (**B**) Multiple alignments of amino acid sequences of PGRP-S in different species. Conserved Zn^2+^-binding amino acids that are essential for amidase activity of PGRPs are indicated in the red triangle.

**Figure 3 ijms-23-11062-f003:**
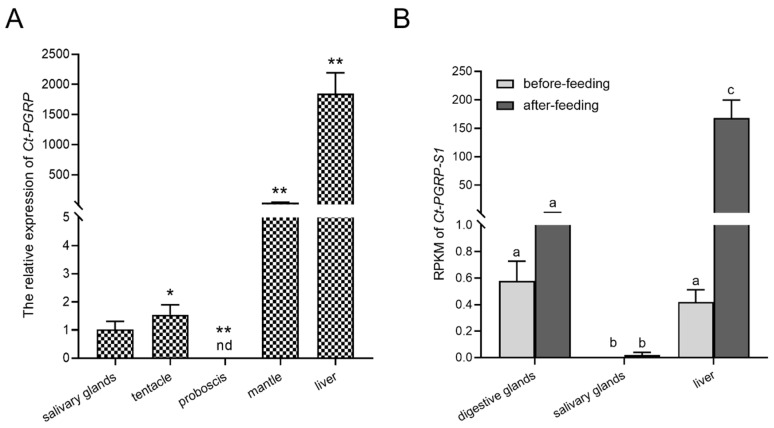
(**A**) Tissue distribution of *Ct-PGRP-S1* mRNA. The selected tissues included the liver, proboscis, mantle, tentacle and salivary glands. The data are presented as the means ± SD (*n* ≥ 3); significant differences between different groups are indicated with * at *p* < 0.05, and with ** at *p* < 0.01; “nd” indicates not detected. (**B**) RPKM of *Ct-PGRP-S1* mRNA in different tissues before and after feeding. Different letters denote significant differences.

**Figure 4 ijms-23-11062-f004:**
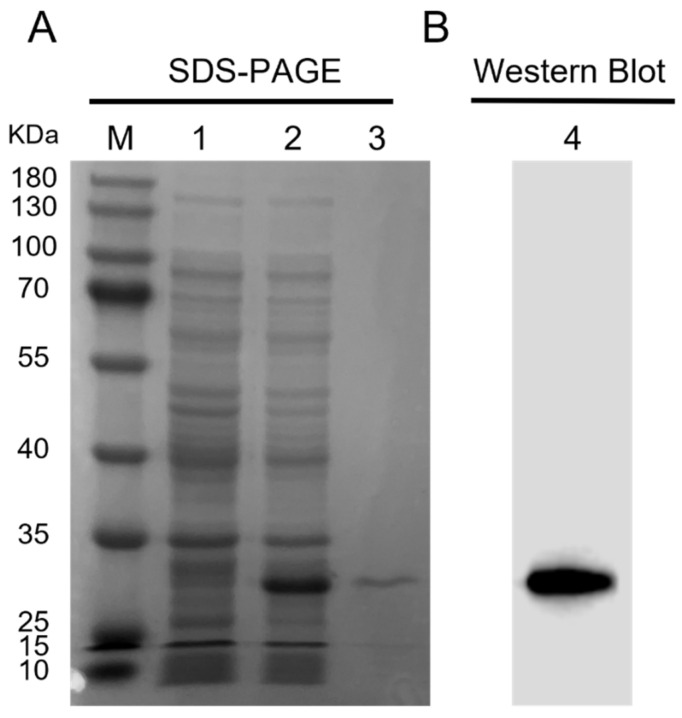
Prokaryotic expression and Western blot analysis of rCt-PGRP-S1. (**A**) The SDS-PAGE analysis of prokaryotic expressed rCt-PGRP-S1. Lane M: protein marker; 1 and 2: protein from BL21 (DE3) transformed with pET28b-rCt-PGRP-S1 plasmid before and after 0.5 mM IPTG induction; 3: purified rCt-PGRP-S1. (**B**) Western blot analysis of rCt-PGRP-S1 with anti-6×His tag monoclonal antibody.

**Figure 5 ijms-23-11062-f005:**
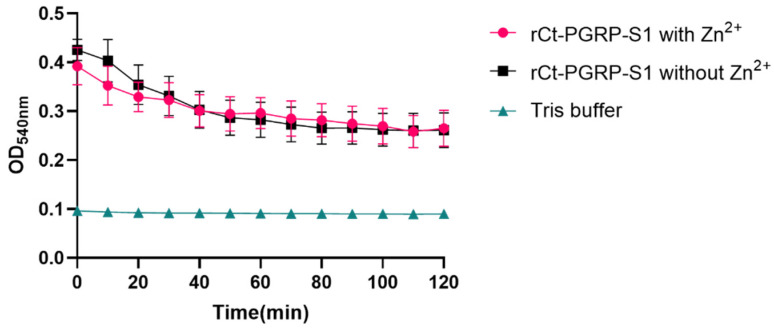
Activity of rCt-PGRP-S1 in the degradation of PGN. rCt-PGRP-S1 was incubated with PGN with or without Zn^2+^, and Tris buffer without rCt-PGRP-S1 and PGN was used as a control. The OD value was recorded at 540 nm per 10 min, lasting for 120 min. For growth curves, biological replicates (*n* = 7) are shown as points with their average values connected by lines. Error bars indicate the standard error of the mean (SEM).

**Figure 6 ijms-23-11062-f006:**
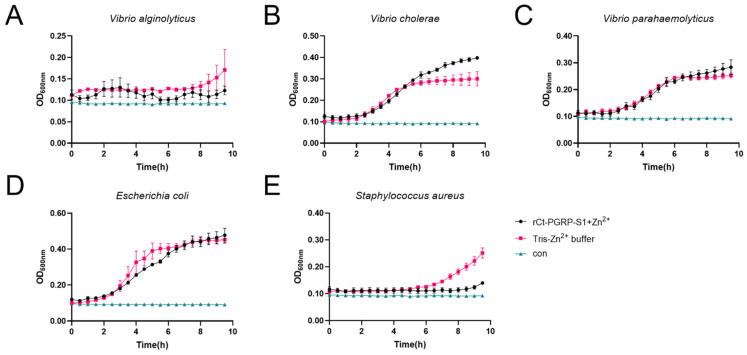
Growth suppressive test of rCt-PGRP-S1 against bacteria. *V. alginolyticus* (**A**), *V. cholerae* (**B**) *V. parahaemolyticus* (**C**), *E. coli* (**D**), and *S. aureus* (**E**) were mixed with rCt-PGRP-S1 (final concentration 50 μg/mL), and the OD_600nm_ was recorded per half hour lasting for 10 h. The wells in which the recombinant protein was replaced with Tris were used as the loading control. The wells in which the recombinant protein and the bacterial culture were left out were used as the blank control (termed con). For growth curves, three biological replicates are shown as points with their average values connected by lines. Error bars indicate the standard error of the mean (SEM).

**Table 1 ijms-23-11062-t001:** Nucleotide sequences of primers used in this study.

Primers	Sequence (5′-3′)
For ORF cloning	
*Ct-PGRP-S1*-F	ATGCATCTGGCCATCATTCTG
*Ct-PGRP-S1*-R	GGTACCATAACGATGCAACG
For qPCR	
Q*Ct-PGRP-S1*-F	CAGTGGCAAGTTCTCTGCAG
Q*Ct-PGRP-S1*-R	CTCTCACCAATAACTGCGCC
Q18S-F	ATGGTCAGAACTACGACGGTAT
Q18S-R	GTATTGCGGTGTTAGAGGTGAA
For recombinant Ct-PGRP-S1 protein construction	
pET28b-F	CACCACCACCACCACCAC
pET28b-R	GGTATATCTCCTTCTTAAAGTTAAACAAAATTATTTC
Ct-PGRP-S1-orf_F	ctttaagaaggagatataccATGCATCTGGCCATCATTC
Ct-PGRP-S1-orf_F	cagtggtggtggtggtggtgCGCACCATTATACAGCGC
pET28b-check-F	AAGTGGCGAGCCCGATCTTC
pET28b-check-R	CTAGGGCGCTGGCAAGTGTA

## Data Availability

The open reading frame (ORF) sequence of *Ct-PGRP-S1* is deposited in GenBank (GenBank ID: OP351736). The other data presented in this study are available in the article.

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
