# Peer review of "Molecular and Functional Characterization of a Short-Type Peptidoglycan Recognition Protein, Ct-PGRP-S1 in the Giant Triton Snail *Charonia tritonis"

_ijms, 2022, doi:10.3390/ijms231911062_

Round 1

Reviewer 1 Report

The manuscript titled “Molecular and functional characterization of a short-type peptidoglycan recognition protein, Ct-PGRP-S1 in the giant triton snail Charonia tritonis” deals with the identification and characterization of the PGRP-S1 protein together with the evaluation of its antibacterial activities suggesting its crucial role in the innate immunity in the endangered species C. tritonis. The research provides interesting insights into the role of this protein in the defence mechanisms of gastropods against pathogens with potential in the mitigation of coral loss.

The methodological approaches involve a wide range of expertise; however, the manuscript needs some work to clarify specific aspects in almost all the sections and there is still a large margin of improvement. I would invite the authors to read carefully the ms before publication; spacing and lack of uniformity denote scarce attention in the presentation of their research (i.e. space before reference number, italics in “Ct-PGRP-S1”, please check throughout the ms, including figures and figure legends).

The Introduction section gives sufficient background to highlight the novelty of the study. However, the incipit is not invited reading as the authors suppose that the topic is familiar to most people. The aim of the study is well-defined, but the findings are not clearly reported (lines 80-92). The authors point out that PGRPs in marine organisms/invertebrates (please avoid “marine PGRPs”) have crucial roles in the immune system and emphasize the importance to investigate the immune response in the endangered species C. tritonis but it’s not clear why they identified six genes and characterize just one cDNA sequence (“Ct-PGRP-S1 (Unigene31272_All)”), why they evaluated the expression level of “Ct-PGRP-S (gene ID: CL2507.Contig2_All and Unigene31272_All)”; probably “Ct-PGRP-S” is Ct-PGRP-S1, but anyway I cannot understand why they mention the two contigs (?) in this section. I suggest rephrasing the entire period, also avoiding “its” before “cDNA” and “mRNA” “of Ct-PGRP-S1”. One of the most serious flows of the manuscript is the complete absence of the accession numbers of the nucleotide and protein sequences used in the study. The authors can state that the data related to the transcriptome of the giant triton snail (Charonia tritonis) are “not published data”, but the nucleotide sequence newly identified should be submitted and deposited to any public genetic sequence databases [DNA DataBank of Japan (DDBJ), the European Nucleotide Archive (ENA), GenBank at NCBI] together with the relative predicted putative protein sequence. Details should be reported in Materials and Methods, together with the accession numbers of the sequences used for the phylogenetic analysis.

The Results section needs to be improved as well. The incipit of the paragraph 2.1 “Based on this nucleotide sequence from transcriptomics” is nonsense: please explain “this” and modulate the paragraph following a chronological order such as (1) identification of Ct-PGRP cDNA sequence from transcriptome of C. tritonis, (2) its molecular characterization including putative deduced protein sequence and bioinformatic analyses for protein domains and 3-D structure, (3) blast search and phylogenetic analysis, (4) tissue distribution and feeding trial, (5) cloning, purification of the recombinant protein and functional assays.

Blast search results must report the accession numbers of the sequences just after the name of the species. Phylogenetic analysis is a relevant tool to identify the genetic distances among species but the tree without the support values at the nodes and the scale bar is meaningless; moreover, the presence of Escherichia coli in the tree is not comprehensible. Please, select the sequences used for this analysis and complete the tree adding the support values at the nodes and the scale bar. All the sequences used “to study the evolutionary relationships by phylogenetic analysis” should be identified by accession numbers and must be reported somewhere, even in supplementary material, or tables/figure; please fix “range of …genes” (line 108).

Please check the panels A and B in Fig. 2 that are inverted in the figure legend; actually, the legend is a mess of misleading information:

-          Ct-PGRP dimer”, should probably be “Ct-PGRP-S1” but the authors never report on dimer, please explain. The figure does not show a dimer.

-          “Structural domain of Lv-AE3”: what is this? Is the same protein? The same species? The same domain?

The panel C in the same figure should show the “Amino acid sequences alignment of 31272 in multiple species” but the authors represent a phylogenetic tree beside the alignment of Peptidoglycan recognition proteins (probably?) with marked Amidase_2 and SH3_3 domains once again (Fig. 1 and Fig. 2 panels A-C). The legend either the figure should be modified to avoid redundancy and the authors should state clearly if the aligned sequences are coding for PGRPs or PGRP-S proteins. Are the sequences used for the phylogenetic analysis coding for short proteins similar to the identified Ct-PGRP-S1? “31272” is unique in the transcriptome found by the authors and should be substituted by a conventional way for sequence identification; no one of the “multiple species” (I would suggest “different species”) has 31272 gene!

Regarding the tissue distribution of Ct-PGRP-S1 mRNA (panel A, Fig. 3) is not clear why the authors did not report the expression in the digestive gland that instead appear in the panel B of the same figure. Did they evaluate the tissue distribution of Ct-PGRP-S1 mRNA in the digestive gland? Why is it not shown? What does it mean that the expression of “Ct-PGRP-S1 was in lower level in the liver of C. tritonis”, lower compared to what? Please explain better that “feeding had no effect on the expression of Ct-PGRP-S1 in the digestive” and salivary glands. Indeed, the statistical data are not clearly reported; apart the significance levels, it is not clear the groups analysed for eventual differences; “between different groups” (panel A, tissue distribution) could significate that the level of expression in the liver are significantly different from that in the salivary glands but the double asterisks mark liver, mantle and also proboscis, but it’s clear the levels of expression are different. Somewhat confusing is the statistics in the comparison before- and after-feeding, please fix it and clearly define the groups that are compared in each case.

In the paragraph 2.3, the inferred amidase activity was speculated by domain prediction, but please note that the “high sequence similarity” (line 150) is not reported neither as % similarity, nor as support values at the tree nodes.

The Discussion section should be improved for the phylogenetic data as the “separate clade” should be clearly defined; separated from what? Other gastropods? Please clarify.

The conserved Zn2+-binding residues in the PGRP domain of Ct-PGRP-S1 are four but in this section the authors discuss three residues: which ones? Some period must be modified (lines 221-223) as these lines report the exact abstract of the cited reference; this is not admissible.

The Materials and Methods section needs to be rearranged: molecular cloning should be a step following animal dissection for transcriptomic database construction; I suggest inverting the first two paragraphs.

These suggestions can improve the strength of the obtained results and the presentation of the novel findings of this study.

Minor points

Please check throughout the text: the spaces before reference number (e.g. lines 44, 46, 47, 58, 59, 60, 65, 68, 70, etc); italics in “Ct-PGRP-S1” (e.g. lines 21, 22, 24, 25, etc) or choose one way to be consistent throughout the ms, including figures and figure legends.

Italics for species’ name should be fixed throughout the text and reference list

Lines 20-21: should probably be “regarding its potential function in the giant triton snail”.

Line 26: PGN is not defined yet; moreover in “non-Zn2+-dependent amidase activity” please fix the apex.

Line 55: please delete underscore

Line 60: fix the references “phagocytosis[1] [18-20]”

Line 61: “these” should be “These”

Lane 74: “of C. tritonis” should be “in C. tritonis

Lines 76-79: I would suggest to merge the two sentences to reduce redundancy.

Lines 80-85: “which play most vital roles in antibacterial immune response in C. tritonis” is not clear. The authors report the identification of six PGRP genes, but they analysed the expression levels of just one of them. Please read the comment above.

Line 85: please delete “detailedly” and please fix “their” since you characterize only one gene.

Line 87: “Ct-PGRP-S1” is italicized, please choose one way and be consistent throughout the ms, including figures.

Lines 89-90: please delete “assay”

Line 97: 288-a.a., line 104: 22-aa, please choose one way and be consistent throughout the ms

Line 122: please substitute “could be” with “was”

Lines 126, 128: please fix the italics

Line 151: please substitute “has” with “could have”

Line 157: OD540nm, check the OD value and indicate them consistently throughout the ms; delete full stop after parenthesis

Line 168: “V. Cholerae” should be fixed throughout the ms as V. cholera

Line 186: please substitute “an” with “the”

Line 187-188: please define “a typical PGRP domain”

Line 255: “separated” should be “collected”

Line 259: “followed the” please substitute with “following”

Lines 262-263: sentence is truncated, without any verb

Line 275: please indicate the biological and technical replicates

Line 278: should be “coding”; add “bp” after 864

Line 279: please substitute “were” with “was”

Line 288: “°C” here is following the number after a space but is not uniform in the text; please, check for it and “xg” and indicate them consistently throughout the ms

Line 381: “stored in PBS buffer and stored in -80°C” substitute with “stored in PBS buffer at -80°C”

Line 309: ZnCl2, please fix the subscript; please clarify the sentence in lines 309-310

Line 314: reference “Jia et al” is missing

In the Reference list the name of the specie must be italicized; ref 51 is all capital letter, please fix it

Figures

Fig. 2: the legend is a mess of misleading information. Please read the comments above.

Figure 3: please order the column data according to the legend (panel B) and explain why the expression levels in “Digestive glands” were not investigated for tissue distribution (panel A). Please check the legend for “growth curves”

Figure 4: fix the legend by capital letter in “Prokaryotic”, “Lane M”

Figure 6: in the legend fix italics, name of the species and “was recorded”

Author Response

Dear reviewers:

Re: Manuscript ID: ijms-1895663 and Title: Molecular and functional characterization of a short-type peptidoglycan recognition protein, Ct-PGRP-S1 in the giant triton snail Charonia tritonis.

Thank you for your letter and the reviewers’ comments concerning our manuscript entitled “Molecular and functional characterization of a short-type peptidoglycan recognition protein, Ct-PGRP-S1 in the giant triton snail Charonia tritonis” (ijms-1895663). Those comments are valuable and very helpful. We have read through comments carefully and have made corrections. Based on the instructions provided in your letter, we uploaded the file of the revised manuscript. Revisions in the text are shown using Yellow highlight. The responses to the reviewer's comments are marked in red and presented following.

We would love to thank you for allowing us to resubmit a revised copy of the manuscript and we highly appreciate your time and consideration. Should you have any questions, please contact us without hesitation.

Sincerely.

Bing LIU

Reviewer #1:

Comments and Suggestions for Authors

The manuscript titled “Molecular and functional characterization of a short-type peptidoglycan recognition protein, Ct-PGRP-S1 in the giant triton snail Charonia tritonis” deals with the identification and characterization of the PGRP-S1 protein together with the evaluation of its antibacterial activities suggesting its crucial role in the innate immunity in the endangered species C. tritonis. The research provides interesting insights into the role of this protein in the defence mechanisms of gastropods against pathogens with potential in the mitigation of coral loss.

The methodological approaches involve a wide range of expertise; however, the manuscript needs some work to clarify specific aspects in almost all the sections and there is still a large margin of improvement. I would invite the authors to read carefully the ms before publication; spacing and lack of uniformity denote scarce attention in the presentation of their research (i.e. space before reference number, italics in “Ct-PGRP-S1”, please check throughout the ms, including figures and figure legends).

Response 1: We are grateful for the suggestion and we realized this misuse of the word knockdown after receiving your opinion. We carefully check the writing of “Ct-PGRP-S1” throughout the whole manuscript and we modified them: italics in “Ct-PGRP-S1” means gene name, “Ct-PGRP-S1” means its coding protein. And we will pay more attention to the consistency of spacing and formatting of the papers.

The Introduction section gives sufficient background to highlight the novelty of the study. However, the incipit is not invited reading as the authors suppose that the topic is familiar to most people. The aim of the study is well-defined, but the findings are not clearly reported (lines 80-92). The authors point out that PGRPs in marine organisms/invertebrates (please avoid “marine PGRPs”) have crucial roles in the immune system and emphasize the importance to investigate the immune response in the endangered species C. tritonis but it’s not clear why they identified six genes and characterize just one cDNA sequence (“Ct-PGRP-S1 (Unigene31272_All)”), why they evaluated the expression level of “Ct-PGRP-S (gene ID: CL2507.Contig2_All and Unigene31272_All)”; probably “Ct-PGRP-S” is Ct-PGRP-S1, but anyway I cannot understand why they mention the two contigs (?) in this section. I suggest rephrasing the entire period, also avoiding “its” before “cDNA” and “mRNA” “of Ct-PGRP-S1”. One of the most serious flows of the manuscript is the complete absence of the accession numbers of the nucleotide and protein sequences used in the study. The authors can state that the data related to the transcriptome of the giant triton snail (Charonia tritonis) are “not published data”, but the nucleotide sequence newly identified should be submitted and deposited to any public genetic sequence databases [DNA DataBank of Japan (DDBJ), the European Nucleotide Archive (ENA), GenBank at NCBI] together with the relative predicted putative protein sequence. Details should be reported in Materials and Methods, together with the accession numbers of the sequences used for the phylogenetic analysis.

Response 2: We apologized for the confusion generated by the previous version of the manuscript. We highly agreed the suggestion about the logic flow in the last paragraph of introduction section and We have modified it. We uploaded the nucleotide sequence of Ct-PGRP-S1 to NCBI GenBank and the accession ID was shown both in MS and Data Availability Statement and we also added the sample ID of the transcript data of Charonia tritonis in the new version. The accession numbers of the sequences used for the phylogenetic analysis was also shown in MS section.

The Results section needs to be improved as well. The incipit of the paragraph 2.1 “Based on this nucleotide sequence from transcriptomics” is nonsense: please explain “this” and modulate the paragraph following a chronological order such as (1) identification of Ct-PGRP cDNA sequence from transcriptome of C. tritonis, (2) its molecular characterization including putative deduced protein sequence and bioinformatic analyses for protein domains and 3-D structure, (3) blast search and phylogenetic analysis, (4) tissue distribution and feeding trial, (5) cloning, purification of the recombinant protein and functional assays.

Response 3: We feel sorry about that. We first obtained the transcript sequence in the transcriptome, and based on this, we designed primers to confirm the sequence of open reading frame of Ct-PGRP-S1. We will improve it. Besides, we have modulated the paragraph following a chronological order based on your suggestion.

Blast search results must report the accession numbers of the sequences just after the name of the species. Phylogenetic analysis is a relevant tool to identify the genetic distances among species but the tree without the support values at the nodes and the scale bar is meaningless; moreover, the presence of Escherichia coli in the tree is not comprehensible. Please, select the sequences used for this analysis and complete the tree adding the support values at the nodes and the scale bar. All the sequences used “to study the evolutionary relationships by phylogenetic analysis” should be identified by accession numbers and must be reported somewhere, even in supplementary material, or tables/figure; please fix “range of …genes” (line 108).

Response 4: Thank you for your advice and We feel sorry about that. The accession numbers of the sequences used for the phylogenetic analysis was shown in figure and MS section. We have improved our phylogenetic tree in the new version.

Please check the panels A and B in Fig. 2 that are inverted in the figure legend; actually, the legend is a mess of misleading information:

Response 5: We feel sorry about that. We checked carefully and modified them based on your suggestion.

 “Ct-PGRP dimer”, should probably be “Ct-PGRP-S1” but the authors never report on dimer, please explain. The figure does not show a dimer.

“Structural domain of Lv-AE3”: what is this? Is the same protein? The same species? The same domain?

Response 6: Thanks for the reminder, yes, we did not report on dimer. The dimer and Lv-AE3 were typos. Sorry, we have made improvements.

The panel C in the same figure should show the “Amino acid sequences alignment of 31272 in multiple species” but the authors represent a phylogenetic tree beside the alignment of Peptidoglycan recognition proteins (probably?) with marked Amidase_2 and SH3_3 domains once again (Fig. 1 and Fig. 2 panels A-C). The legend either the figure should be modified to avoid redundancy and the authors should state clearly if the aligned sequences are coding for PGRPs or PGRP-S proteins. Are the sequences used for the phylogenetic analysis coding for short proteins similar to the identified Ct-PGRP-S1? “31272” is unique in the transcriptome found by the authors and should be substituted by a conventional way for sequence identification; no one of the “multiple species” (I would suggest “different species”) has 31272 gene!

Response 7: Thank you for your helpful suggestions. We have improved it, which can be seen in figure 1 and figure 2 in the new version.

Regarding the tissue distribution of Ct-PGRP-S1 mRNA (panel A, Fig. 3) is not clear why the authors did not report the expression in the digestive gland that instead appear in the panel B of the same figure. Did they evaluate the tissue distribution of Ct-PGRP-S1 mRNA in the digestive gland? Why is it not shown? What does it mean that the expression of “Ct-PGRP-S1 was in lower level in the liver of C. tritonis”, lower compared to what? Please explain better that “feeding had no effect on the expression of Ct-PGRP-S1 in the digestive” and salivary glands. Indeed, the statistical data are not clearly reported; apart the significance levels, it is not clear the groups analysed for eventual differences; “between different groups” (panel A, tissue distribution) could significate that the level of expression in the liver are significantly different from that in the salivary glands but the double asterisks mark liver, mantle and also proboscis, but it’s clear the levels of expression are different. Somewhat confusing is the statistics in the comparison before- and after-feeding, please fix it and clearly define the groups that are compared in each case.

Response 8: Thank you for your helpful suggestion. Because of the limited sample, we did not do the tissue distribution of digestive glands. We have redone statistics in the comparison before- and after-feeding, it may be more clearly to see the differences between groups. We also reanalyzed this data.

In the paragraph 2.3, the inferred amidase activity was speculated by domain prediction, but please note that the “high sequence similarity” (line 150) is not reported neither as % similarity, nor as support values at the tree nodes.

Response 9: Thank you for your suggestion. We have improved the figure 2 and show the values at the tree nodes.

The Discussion section should be improved for the phylogenetic data as the “separate clade” should be clearly defined; separated from what? Other gastropods? Please clarify.

Response 10: Thank you for your helpful suggestion and we have improved this discussion.

The conserved Zn2+-binding residues in the PGRP domain of Ct-PGRP-S1 are four but in this section the authors discuss three residues: which ones? Some period must be modified (lines 221-223) as these lines report the exact abstract of the cited reference; this is not admissible.

Response 11: thank you for your suggestion. We have revised the figure 2 in the new version. We have improved the content (new version in line 209-212).

The Materials and Methods section needs to be rearranged: molecular cloning should be a step following animal dissection for transcriptomic database construction; I suggest inverting the first two paragraphs.

Response 12: thank you for your helpful suggestion and we have modified this.

These suggestions can improve the strength of the obtained results and the presentation of the novel findings of this study.

Minor points

Please check throughout the text: the spaces before reference number (e.g. lines 44, 46, 47, 58, 59, 60, 65, 68, 70, etc); italics in “Ct-PGRP-S1” (e.g. lines 21, 22, 24, 25, etc) or choose one way to be consistent throughout the ms, including figures and figure legends.

Response 13: We are grateful for the suggestion and we carefully proof-read the manuscript and revised them.

Italics for species’ name should be fixed throughout the text and reference list

Response 14: We have revised them.

Lines 20-21: should probably be “regarding its potential function in the giant triton snail”.

Response 15: We have revised them (line 20-21).

Line 26: PGN is not defined yet; moreover in “non-Zn2+-dependent amidase activity” please fix the apex.

Response 16: We have revised them.

Line 55: please delete underscore

Line 60: fix the references “phagocytosis[1] [18-20]”

Response 17: We have revised them.

Line 61: “these” should be “These”

Lane 74: “of C. tritonis” should be “in C. tritonis

Response 18: We have revised them.

Lines 76-79: I would suggest to merge the two sentences to reduce redundancy.

Response 19: we are grateful for this suggestion. We have revised them.

Lines 80-85: “which play most vital roles in antibacterial immune response in C. tritonis” is not clear. The authors report the identification of six PGRP genes, but they analysed the expression levels of just one of them. Please read the comment above.

Response 20: We have revised them (line 80-81).

Line 85: please delete “detailedly” and please fix “their” since you characterize only one gene.

Line 87: “Ct-PGRP-S1” is italicized, please choose one way and be consistent throughout the ms, including figures.

Response 21: We have revised them. We choose this way to express our exact meaning: Gene name is showed italicized but protein should be not.

Lines 89-90: please delete “assay”

Line 97: 288-a.a., line 104: 22-aa, please choose one way and be consistent throughout the ms

Response 22: We have revised them.

Line 122: please substitute “could be” with “was”

Lines 126, 128: please fix the italics

Response 23: We have revised them.

Line 151: please substitute “has” with “could have”

Line 157: OD540nm, check the OD value and indicate them consistently throughout the ms; delete full stop after parenthesis

Response 24: We have revised them.

Line 168: “V. Cholerae” should be fixed throughout the ms as V. cholera

Response 25: We have revised them.

Line 186: please substitute “an” with “the”

Response 26: We have revised them.

Line 187-188: please define “a typical PGRP domain”

Response 27: We have revised them.

Line 255: “separated” should be “collected”

Response 28: We have revised them.

Line 259: “followed the” please substitute with “following”

Response 29: We have revised them.

Lines 262-263: sentence is truncated, without any verb

Response 30: We have revised them.

Line 275: please indicate the biological and technical replicates

Response 31: We have revised them.

Line 278: should be “coding”; add “bp” after 864

Line 279: please substitute “were” with “was”

Response 32: We have revised them.

Line 288: “°C” here is following the number after a space but is not uniform in the text; please, check for it and “xg” and indicate them consistently throughout the ms

Response 33: We have revised them.

Line 381: “stored in PBS buffer and stored in -80°C” substitute with “stored in PBS buffer at -80°C”

Response 33: We have revised it.

Line 309: ZnCl2, please fix the subscript; please clarify the sentence in lines 309-310

Line 314: reference “Jia et al” is missing

Response 34: We have revised them.

In the Reference list the name of the specie must be italicized; ref 51 is all capital letter, please fix it

Response 35: We have revised them.

Figures

Fig. 2: the legend is a mess of misleading information. Please read the comments above.

Response 36: We have revised them.

Figure 3: please order the column data according to the legend (panel B) and explain why the expression levels in “Digestive glands” were not investigated for tissue distribution (panel A). Please check the legend for “growth curves”

Response 37: We have revised them.

Figure 4: fix the legend by capital letter in “Prokaryotic”, “Lane M”

Figure 6: in the legend fix italics, name of the species and “was recorded”

Response 38: We have revised them.

Reviewer 2 Report

Please see the attached file for comments.

Author Response

Dear reviewers:

Re: Manuscript ID: ijms-1895663 and Title: Molecular and functional characterization of a short-type peptidoglycan recognition protein, Ct-PGRP-S1 in the giant triton snail Charonia tritonis.

Thank you for your letter and the reviewers’ comments concerning our manuscript entitled “Molecular and functional characterization of a short-type peptidoglycan recognition protein, Ct-PGRP-S1 in the giant triton snail Charonia tritonis” (ijms-1895663). Those comments are valuable and very helpful. We have read through comments carefully and have made corrections. Based on the instructions provided in your letter, we uploaded the file of the revised manuscript. Revisions in the text are shown using Yellow highlight. The responses to the reviewer's comments are marked in red and presented following.

We would love to thank you for allowing us to resubmit a revised copy of the manuscript and we highly appreciate your time and consideration. Should you have any questions, please contact us without hesitation.

Sincerely.

Bing LIU

The manuscript by Liu et al is the first report on a short-type peptidoglycan recognition protein, Ct-PGRP-S1 in the giant triton snail (Charonia tritonis).

  1. tritonis, an endangered gastropod species of ecological and economic importance, is widely distributed in coral reef ecosystems of the Indo-West Pacific region and the tropical waters of the South China Sea. The predatory giant triton is an echinoderm specialist with a preference for asteroids including the crown-of-thorns starfish (Acanthaster planci), which is reported to be the primary cause of coral cover loss on many reefs. Aquaculture is considered one approach that could help restore natural populations of C. tritonis and mitigate coral loss. However, bacterial infections in aquaculture systems cause a huge loss in productivity and remain a major challenge for the growth of the aquaculture industry. The Authors suggest the importance of peptidoglycan recognition protein Ct-PGRP-S1 in the antibacterial defense of C. tritonis.

Although the subject matter is interesting, some important concerns need to be addressed before the manuscript is ready for publication in IJMS.

General comments:

  1. The presentation of results should be improved. Not all conclusions are sufficiently supported by the obtained results. The Materials and Methods section is sometimes unclear.

Response 1: Thank you for your helpful suggestion. We have improved the results and modified some less supportive conclusions based on your suggestion.

Specific comments:

  1. Lines 20-21: I suggest to correct the sentence “However, …..”.

Response 2: We are grateful for the suggestion and we have revised this sentence (Lines 20-21).

  1. Line 26: Acronyms should be defined the first time they appear in the text. I suggest “to degrade peptidoglycan (PGN)” or “to degrade peptidoglycan” instead of “to degrade PGN”. See also line 40 for PGN.

Response 3: We are grateful for the suggestion and we have revised it.

  1. Line 26: Superscript is necessary. It should be “non-Zn2+-dependent” instead of “non-Zn2+-dependent”.

Response 4: We are grateful for the suggestion and we have revised it.

  1. Lines 44, 46, 47, 58-60, 65, 68, 70, 152, 209, 217, 223: There should be a space before the reference numbers.

Response 5: Thank you for your suggestion. We checked carefully throughout the whole manuscript and modified them based on your suggestion.

  1. Line 46: The acronym is incorrect. It should be “PGRP” instead of “PGPR”.

Response 6: we have revised it.

  1. Line 55: It should be “defense.” instead of “defense._”.

Response 7: we have revised it (line 56).

  1. Line 60: It should be “[1, 18-20]” instead of “[1] [18-20]“.

Response 8: we have revised it (line 61).

  1. Line 61: It should be “ These” instead of “these“.

Response 9: we have revised it (line 62).

  1. Lines 80-81: The sentence is not clear enough.

Response 10: we have revised it (line 79-80).

  1. Lines 90-92: The statement is not sufficiently supported by the obtained results.

Response 11: we have improved this statement (line 91-95).

  1. Lines 95-96: It should be “open reading frame (ORF) of Ct-PGRP ” instead of “open reading frame of Ct-PGRP“. See also line 97 for ORF and comment 3.

Response 12: we have improved it (line 101).

  1. Lines 97-98: It should be “288-aa protein precursor” instead of “288-a.a. protein precursor“. Please see also line 104.

Response 13: we have improved it (line 103).

  1. Page 3, Figure 1: Not all conserved Zn2+-binding amino acids that are essential for amidase activity of PGRPs (in red) are consistent with those listed in lines 209-211.

Response 14: We are grateful for the reminder. We have improved it (Figure 2).

  1. Lines 104-107: The statements are not consistent with Figure 2A and 2B captions. Please see comment 16.

Response 15: We are grateful for the reminder. We have revised it (line 108-110).

  1. Page 4, Figure 2: Figure 2A and 2B captions should be corrected to be consistent with panel A and panel B of Figure 2. Please see also comment 15.

Response 16: We are grateful for the reminder. We have revised the figure 2 (new version in figure 1).

  1. Line 126: I suggest “was seen” instead of “was saw”. Italics should not be used.

Response 17: We have revised it (line 141)

  1. Line 128: It should be “was in lower level in the liver of” instead of “was in lower level in the liver of“. Italics should not be used.

Response 18: We have revised it (line 146).

  1. Page 5, Fig. 3: Figure 3 should be corrected. Indications of significant differences should be more precisely placed. It is not easy to understand “nd” with two asterisks in panel B of Figure 3.

Response 19: Thank you for your advice. We have added this indication of “nd” in the caption of figure 3 (line 152).

  1. Line 145 (Figure 4 caption): It should be “Prokaryotic expression and Western blot” instead of “prokaryotic expression and western blot”.

Response 20: Thank you for your advice. We have revised it (line 162).

  1. Line 146: It should be “ Lane M: protein marker” instead of “Lan m: protein marker“.

Response 21: Thank you for your advice. We have revised it (line 163).

  1. Line 147: It should be “pET-28b-rCt-PGRP-S1 ” instead of “pET-28b- rCt-PGRP-S1“ and “0.5 mM” instead of “0.5mM”.

Response 22: Thank you for your advice. We have revised it (line 164).

  1. Line 148: It should be “anti-6×His” instead of “anti- 6×His“.

Response 23: Thank you for your advice. We have revised it (line 165).

  1. Lines 154 and 308: It should be “Tris-ZnCl2” instead of “Tris-ZnCl2“. Subscript should be used.

Response 24: Thank you for your advice. We have revised them (line 172 and 348).

  1. Line 156: “Zn2+ group” - the term “group” in relation to ions is incorrect.

Response 25: Thank you for your advice. We have revised it (line 174).

  1. Lines 155-159: It should be one sentence (there should be no period after the parentheses).

Response 26: Thank you for your advice. We have revised it (line 176-178).

  1. Lines 159-160: The sentence is not clear enough. The term “rCt-PGRP-S1” is not consistent with Refs. 16 and 21.

Response 27: Thank you for your advice. We have revised it (line 178-179).

  1. Lines 162-165, Figure 5 caption: Control should be described.

Response 28: Thank you for your advice. We have revised it (line 181-183).

  1. Lines 168, 172, 177, 322: It should be “V. cholerae ” instead of “V. Cholerae“.

Response 29: Thank you for your advice. We have revised them (line 187, 189, 196, 356, 362).

  1. Lines 171-173: The sentence should be corrected. Data presented for V. cholerae (Fig. 6, panel B) and V. parahaemolyticus (Fig. 6, panel C) do not support the inhibition by rCt-PGRP-S1. Moreover, I suggest to correct “merely and weakly”.

Response 30: Thank you for your advice. We have improved it (line 190-191).

  1. Line 172: It should be “as they” instead of “as they“. Italics should not be used.

Response 31: Thank you for your advice. We have improved it (line 191).

  1. Page 7, Figure 6: Legend should be more consistent with the Subsection 4.7. of Materials and Methods. Please see also comment 33.

Response 32: Thank you for your advice. We have improved them (figure 6, 356).

  1. Lines 177-181, Figure 6 caption: Control should be described. Please see also comment 32.

Response 33: Thank you for your advice. We have improved them (line 198-201).

  1. Line 178: : It should be “were mixed with” instead of “were mixed with “. Italics should not be used.

Response 34: Thank you for your advice. We have revised it (line 197).

  1. Line 179: The fragment about the OD600nm should be corrected.

Response 35: Thank you for your advice. We have revised it (line 198).

  1. Lines 209-211: Please see comment 14 for conserved Zn2+-binding residues.

Response 36: We are grateful for the reminder. We have improved it (figure 2).

  1. Line 210: I suggest “Ct-PGRP-S1 has all” instead of “Ct-PGRP-S1 have all”.

Response 37: Thank you for your advice. We have improved it (line 235).

  1. Lines 228-231: The statement about antimicrobial activity against Vibrio is not supported by the results presented for V. cholerae (Fig. 6, panel B) and V. parahaemolyticus (Fig. 6, panel C). Please see also comment 30.

Response 38: Thank you for your advice. We have improved it (line 252-254).

  1. Lines 235-237: The conclusion about antimicrobial activity against Vibrio is not supported by the results. Please see also comments 38 and 30.

Response 39: Thank you for your advice. We have improved it (line 262).

  1. Lines 258-260: The sentence is repeated and should be deleted.

Response 40: Thank you for your advice. We have improved it (line 271-274).

  1. Line 288: I suggest “50 μg of kanamycin” instead of “50 μg Kanamycin”.

Response 41: Thank you for your advice. We have improved it (line 323).

  1. Line 295: It should be “50 mM” instead of “50mM”.

Response 42: Thank you for your advice. We have improved it (line 330).

  1. Line 299: It should be “50-500 mM imidazole gradient” instead of “50~500 imidazole gradient”.

Response 43: Thank you for your advice. We have improved it (line 334).

  1. Lines 300-301: It should be “was concentrated using Amicon” instead of “was concentrated used Amicon“.

Response 44: Thank you for your advice. We have improved it (line 336).

  1. Lines 302-303: It should be “sodium dodecyl sulfate-polyacrylamide gel electrophoresis” instead of “sodium dodecyl sulfateepolyacrylamide gel electrophoresis” for SDS-PAGE.

Response 45: Thank you for your advice. We have improved it (line 337-338).

  1. Line 309: It should be “ZnCl2” instead of “ZnCl2“.

Response 46: Thank you for your advice. We have improved it (line 348-349).

  1. Line 313: Ref. [58] concerns the work by He et al. There is no reference to the work by Jia et al.

Response 47: Thank you for your advice. We have improved it (line 354).

  1. Line 316: It should be “cholerae” instead of “Cholerae“.

Response 48: Thank you for your advice. We have improved it (line 356, 362).

  1. Line 316: Vibrio parahaemolyticus is not mentioned. Please see line 322.

Response 49: Thank you for your advice. We have improved it (line 356).

  1. Line 318: I suggest “to investigate” instead of “to investigated”.

Response 49: Thank you for your advice. We have improved it (line 358-359).

  1. Line 320: The term “sterile Tris” is not clear enough (was it 50 mM Tris-HCl, pH 8.0?).

Response 49: Thank you for your advice. We have improved it (line 360).

  1. Lines 321-322: It should be “V. cholerae” instead of “V Cholerae”.

Response 48: Thank you for your advice. We have improved it (line 362).

  1. Lines 328-329: No Supplementary Materials are mentioned in the text of the manuscript.

Response 53: Thank you for your advice. We have deleted it.

  1. Line 343: The sentence should be corrected.

Response 54: Thank you for your advice. We have improved it (line 383).

  1. No italics should be used for recombinant Ct-PGRP-S1 protein throughout the manuscript.

Response 55: Thank you for your advice. We checked carefully throughout the manuscript and improved this writing.

  1. References: I suggest using italics for species names.

Response 56: Thank you for your advice. We checked carefully throughout the References and improved them.

  1. Text editing is highly recommended.

Response 57: Thank you for your advice. We will be happy to edit the text further, based on helpful comments from the reviewers.

Round 2

Reviewer 1 Report

Dear Authors,

I congratulate you on the improved manuscript.

I suggest a few further minor points before publication.

Best wishes

Minor points

Lines 80-81: please do not report accession in the introduction.

Lines 123-134: Delete the name of the species in parenthesis. It’s a scientific rule, it is not necessary.

Line 158: please “(Fig. 4A)” consistent with all the ms

Author Response

Dear reviewers:

Re: Manuscript ID: ijms-1895663 and Title: Molecular and functional characterization of a short-type peptidoglycan recognition protein, Ct-PGRP-S1 in the giant triton snail Charonia tritonis.

再次感谢您的来信和审稿人对我们题为“巨型三亚尼共生蜗牛Charonia Triton中的短型肽聚糖识别蛋白Ct-PGRP-S1的分子和功能表征”(ijms-1895663)的手稿的评论。这些评论是有价值的,非常有帮助的。我们仔细阅读了评论并进行了更正。根据您信中提供的说明,我们上传了修订稿的文件。文本中的修订使用红色突出显示显示。对审稿人评论的回复以红色标记,并在下面显示。

我们非常感谢您允许我们重新提交手稿的修订副本,我们非常感谢您的时间和考虑。如果您有任何疑问,请毫不犹豫地与我们联系。

真诚地。

刘兵

Comments and Suggestions for Authors

Dear Authors,

I congratulate you on the improved manuscript.

I suggest a few further minor points before publication.

Best wishes

Minor points

Lines 80-81: please do not report accession in the introduction.

Response 1: Thank you for your advice. We have revised them (line 80-81)

第 123-134 行:删除括号中的物种名称。这是一个科学规则,没有必要。

回应2:感谢您的有用建议。我们修订了它们(第126行)

第158行:请“(图4A)”与所有ms一致

回应3:感谢您的建议。我们修改了它们(第155-156行),我们也检查了整个手稿。

Reviewer 2 Report

Please see the attached file for comments.

Author Response

Dear reviewers:

Re: Manuscript ID: ijms-1895663 and Title: Molecular and functional characterization of a short-type peptidoglycan recognition protein, Ct-PGRP-S1 in the giant triton snail Charonia tritonis.

Thank you for your letter again and the reviewers’ comments concerning our manuscript entitled “Molecular and functional characterization of a short-type peptidoglycan recognition protein, Ct-PGRP-S1 in the giant triton snail Charonia tritonis” (ijms-1895663). Those comments are valuable and very helpful. We have read through comments carefully and have made corrections. Based on the instructions provided in your letter, we uploaded the file of the revised manuscript. Revisions in the text are shown using Red highlight. The responses to the reviewer's comments are marked in red and presented following.

We would love to thank you for allowing us to resubmit a revised copy of the manuscript and we highly appreciate your time and consideration. Should you have any questions, please contact us without hesitation.

Sincerely

Bing LIU
